# Pathologic Predictors of Response to Treatment of Immune Checkpoint Inhibitor–Induced Kidney Injury

**DOI:** 10.3390/cancers14215267

**Published:** 2022-10-27

**Authors:** Ala Abudayyeh, Liye Suo, Heather Lin, Omar Mamlouk, Noha Abdel-Wahab, Amanda Tchakarov

**Affiliations:** 1Section of Nephrology, Division of Internal Medicine, The University of Texas MD Anderson Cancer Center, 1400 Pressler Street, Houston, TX 77030, USA; 2Department of Pathology and Laboratory Medicine, McGovern Medical School UTHealth, The University of Texas Health Science Center at Houston, 6431 Fannin Street, Houston, TX 77030, USA; 3Department of Biostatistics, The University of Texas MD Anderson Cancer Center, Houston, TX 77030, USA; 4Department of Melanoma Medical Oncology, The University of Texas MD Anderson Cancer Center, Houston, TX 77030, USA; 5Rheumatology and Rehabilitation Department, Faculty of Medicine, Assiut University Hospitals, Assiut 71515, Egypt; 6Section of Rheumatology and Clinical Immunology, Department of General Internal Medicine, The University of Texas MD Anderson Cancer Center, Houston, TX 77030, USA

**Keywords:** immune related adverse event, acute interstitial nephritis, acute kidney injury

## Abstract

**Simple Summary:**

Immune related adverse events (irAEs) are a management challenge with an associated increased morbidity and mortality. The most common renal toxicity is acute interstitial nephritis (AIN), which may be analogous to kidney transplant rejection. In a retrospective cohort of 35 patients with biopsy confirmed AIN, a detailed pathological evaluation was performed using transplant rejection BANFF criteria. The study concluded that patients with increased interstitial fibrosis on kidney biopsies were less likely to have renal response compared to patients with less fibrosis, (*p* = 0.027). Interstitial inflammation, tubulitis, number of eosinophils, neutrophils, and immune subtype of cells had no impact on renal response. In addition, patients who received concurrent ICI and achieved renal response within 3 months had the best OS in comparison to patients who did not receive concurrent ICI nor achieved renal response.

**Abstract:**

Background: Immune-related adverse events are a management challenge in patients receiving immune checkpoint inhibitors (ICIs). The most common renal immune-related adverse event, acute interstitial nephritis (AIN), is associated with patient morbidity and mortality. AIN, characterized by infiltration of renal tissue with immune cells, may be analogous to kidney transplant rejection. We evaluated clinical variables and pathologic findings to identify predictors of renal response and overall survival (OS) in patients with ICI-induced AIN. Design, setting, participants, and measurements: We reviewed the records and biopsy specimens of all 35 patients treated for ICI-induced AIN at our institution, between August 2007 and August 2020, who had biopsy specimens available. Two board-certified renal pathologists graded the severity of inflammation and chronicity using transplant rejection Banff criteria and performed immunohistochemistry analysis. Patients were categorized as renal responders if creatinine had any improvement or returned to baseline within 3 months of initiating treatment for AIN. Clinical and pathologic characteristics and OS were compared between responders and non-responders. Results: Patients with high levels of interstitial fibrosis were less likely to be responders than those with less fibrosis (*p* = 0.02). Inflammation, tubulitis, the number of eosinophils and neutrophils, and the clustering or presence of CD8+, CD4+, CD20+, or CD68+ cells were not associated with renal response. Responders had better OS than non-responders (12-month OS rate 77% compared with 27%, *p* = 0.025). Responders who received concurrent ICIs had the best OS, and non-responders who did not receive concurrent ICIs had the worst OS (12-month OS rate 100% for renal response and concurrent ICIs, 72% for renal response and no concurrent ICIs, and 27% for no renal response and no concurrent ICIs; *p* = 0.041). Conclusions: This is the first analysis of ICI induced nephritis where a detailed pathological and clinical evaluation was performed to predict renal response. Low levels of interstitial fibrosis in kidney tissue are associated with renal response to treatment for ICI-induced AIN, and the renal response and use of concurrent ICIs are associated with better OS in these patients. Our findings highlight the importance of the early diagnosis and treatment of ICI-AIN, while continuing concurrent ICI therapy.

## 1. Introduction

With the widespread use of immune checkpoint inhibitors (ICIs) in cancer treatment, unintended immune responses, termed immune-related adverse events (irAEs), have become an accepted challenge with these novel and life-saving drugs. Identifying predictors and early markers of irAEs has allowed the early intervention and treatment of irAEs without hindering the effectiveness of ICIs. An unintended immune response after ICI exposure can occur in any organ, but irAEs occur most often in the skin, gastrointestinal tract, and endocrine system [1,2].

Renal irAEs occur less frequently, are more difficult to diagnose (in part due to a lack of specific symptoms and diagnostic tests), and are associated with higher patient morbidity and mortality rates compared with other irAEs [1]. ICI-associated acute kidney injury (AKI) may occur in 1.4–16.5% of patients receiving ICI therapy, with median times to AKI diagnosis ranging from 1 to 3 months after ICI exposure [3,4,5,6]. However, data-driven guidance on the use of pathologic findings to treat or assess the prognosis of renal irAEs is lacking. Current societal guidelines for renal irAEs lack a grading system of risk for ICI-associated AKI or a biopsy-related approach to treatment. Although the incidence of ICI-induced AKI is low, ICIs are often used for adjuvant and neoadjuvant treatment of solitary kidney and chronic kidney disease, where progression to dialysis may be imminent, emphasizing the importance of a more informed approach to diagnosis and treatment of renal irAEs [7].

Several researchers have sought to evaluate predictors of AKI and other irAEs after ICI exposure [3,6,8]. Multiple studies have shown that achieving tumor response is associated with irAEs in general [9]. The combined use of anti–cytotoxic T-lymphocyte-associated antigen 4 and anti–programmed cell death protein 1/programmed death-ligand 1 agents has been shown to be an independent predictor of irAEs, including AKI [10,11]. Some authors have reported that low baseline estimated glomerular filtration rates were associated with AKI, but this finding was not consistent across studies [3,11]. Therefore, ICI therapy should not be withheld from patients with impaired kidney function or chronic kidney disease, particularly given the low incidence of AKI in these patients, but providers should nevertheless be aware of how to manage AKI should it arise.

AKI associated with ICIs has been predominately reported to be acute interstitial nephritis (AIN) with severe immune infiltration of the kidney parenchyma, which is often similar to acute cellular kidney transplant rejection. On rare occasions, glomerulonephritis could also be induced by an autoimmune process [12,13,14]. Despite the similarities of ICI-induced AIN to cellular kidney transplant rejection, the use of pathologic findings from biopsy specimens to assess renal irAEs has not been studied. In the current study, we evaluated available biopsy specimens from patients at our institution with ICI-induced AIN for immune cell subtypes, and using the Banff criteria for kidney transplant rejection, we analyzed the association of these characteristics and other clinical variables with renal response to treatment of AIN and survival outcomes.

## 2. Methods

### 2.1. Patient Data Collection

This retrospective study was approved by the Institutional Review Board at The University of Texas MD Anderson Cancer Center, and the procedures followed were in accordance with the principles of the Declaration of Helsinki. We identified patients listed in the MD Anderson pharmacy database who were treated with ipilimumab, nivolumab, pembrolizumab, atezolizumab, durvalumab, or avelumab, between August 2007 and August 2020. Among the 12,195 patients identified, 114 underwent a renal biopsy, and 35 of these had confirmed AIN and tissue specimen slides available for review (Figure 1). The records of these 35 patients were reviewed for baseline demographic characteristics, type of malignancy, comorbidities, number of cycles of ICIs, concurrent nephrotoxic chemotherapies, concurrent use of proton pump inhibitors and nonsteroidal anti-inflammatory drugs, urinalysis findings, the presence of other irAEs, and survival data.

### 2.2. Tissue Evaluation

All hematoxylin and eosin–stained tissue specimen slides were independently and blindly reviewed by two renal pathologists. The severity of inflammation and chronicity were graded using the Banff 2019 Kidney Meeting Report lesion grading system [15]. Quantitative criteria for inflammation (i score) were as follows: 0, no inflammation or inflammation in <10% of unscarred cortical parenchyma; 1, inflammation in 10–25% of unscarred cortical parenchyma; 2, inflammation in 26–50% of unscarred cortical parenchyma; and 3, inflammation in >50% of unscarred cortical parenchyma. Quantitative criteria for tubulitis (t score) were as follows: 0, no mononuclear cells in tubules; 1, foci with one to four cells per tubular cross-section or 10 tubular cells; 2, foci with 5–10 cells per tubular cross-section or 10 tubular cells; and 3, foci with >10 cells per tubular cross-section or the presence of two or more areas of tubular basement membrane destruction accompanied by i2/i3 inflammation and t2 elsewhere. Interstitial fibrosis with tubular atrophy (IFTA) was assessed as a percentage of the affected cortex. In addition, other pathologic findings were recorded: percentage of global glomerulosclerosis, presence of granuloma, and maximum number of interstitial neutrophils and eosinophils per 40× magnification. We also evaluated immune cell subtypes using immunohistochemistry staining for CD4, CD8, CD20, and CD68, for both presence and clustering of cells.

### 2.3. Renal Response and Survival

Upon diagnosis of AIN, all patients received prednisone at a starting dose of 60 mg, and the duration of treatment ranged from 1 week to 12 weeks. Three patients received infliximab in addition to steroids. Creatinine values were evaluated at diagnosis of AIN and at 3 months after initiation of therapy to assess response. The grades of renal toxicity were based on Common Terminology Criteria for Adverse Events: grade 1, creatinine 1.5 times baseline level; grade 2, creatinine >1.5 to 3 times baseline level; grade 3, creatinine >3 to 6 times baseline level; and grade 4, creatinine >6 times baseline level. Patients with any persistent creatinine improvement >0.35 mg/dL at 3 months after initiation of treatment for AIN were considered responders, and those with no improvement in creatinine at that time were considered non-responders.

Progression-free survival (PFS) was defined as the time interval from response evaluation (at 3 months after initiation of treatment for AIN) to progression or death, whichever occurred first. Overall survival (OS) was defined as the time interval from response evaluation to death. For events that had not occurred by the time of data analysis, times were censored at the last contact at which the patient was known to be alive or free of progression.

### 2.4. Statistical Analysis

Descriptive statistics (frequency distribution, mean ± standard deviation, and median and range) were used to summarize patient characteristics. The Fisher exact test was used to compare categorical variables between responders and non-responders, and the *t* test or analysis of variance or their counterparts for nonparametric data (Wilcoxon rank-sum or Kruskal-Wallis test) were used to compare continuous variables between the groups. The distributions of OS and PFS were estimated using the Kaplan-Meier method [16], and the log-rank test [17], was performed to identify differences in survival between groups.

## 3. Results

The characteristics of the 35 patients with confirmed AIN during the period studied are summarized in Table 1. The median follow-up time was 12.2 months, and the median time from ICI therapy initiation to diagnosis of AIN was 123 days. Twenty-two patients (63%) were male with a median age of 67 years at diagnosis of AIN. The most common malignancy was melanoma (n = 10, 29%), followed by lung cancer (n = 7, 20%) and urothelial cancer (n = 3, 9%).

A total of 29 patients (83%) received single-agent immunotherapy; nivolumab (n = 17, 49%) and pembrolizumab (n = 13, 37%) were the most commonly used ICIs (Table 1). Six patients (17%) received two ICIs. The median number of ICI cycles was six (range 1–123 cycles), with a median length of steroid treatment of 4 weeks.

Thirteen patients (37%) were exposed to nonsteroidal anti-inflammatory drugs or proton pump inhibitors. Ten patients (29%) were treated with steroids prior to the kidney biopsy and four of the 10 patients had Banff inflammation scores of 0 or 1. Grade III or IV renal toxicity was noted in 22 (63%) patients; six had to undergo dialysis. 29% of the patients were on concurrent nephrotoxic agents such as carboplatin, pemetrexed, Dabrafenib, trametinib, Sitravatinib, and Axitinib.

Renal toxicity at the time of AIN diagnosis was categorized as grade II in 13 patients (37%), grade III in 10 patients (29%), and grade IV in 12 patients (34%; Table 1). Urinalysis showed that 25 patients (71%) had red or white blood cells present in their urine. Other irAEs were reported in 15 patients (43%); the most common of these were colitis in three patients (9%) and arthritis in two patients (6%).

Kidney biopsy results showed that 31 patients (89%) had chronic interstitial nephritis and 26 (83%) had acute tubular necrosis in addition to AIN (Table 1). In addition, five patients (14%) had granuloma in conjunction with AIN. The median interstitial fibrosis noted was 10% (range 0–70%). The median global glomerulosclerosis observed was 14% (range 0–55%). The median Banff inflammation score was 2 (range 0–3), and the median tubulitis score was 2 (range 0–3). The maximum number of eosinophils observed was 101, with a median of two cells per high-power field. The maximum number of neutrophils observed was 89 cells, with a median of seven cells per high-power field. CD4+, CD8+, CD20+, and CD68+ immune cell subtypes were each observed in more than two-thirds of the biopsy specimens, and CD8+ cells were most often associated with tubulitis (Figure 2).

### 3.1. Renal Response

Twenty-nine patients (83%) achieved complete or partial renal response at 3 months after initiation of treatment for AIN (i.e., responders), and six patients (17%) did not (i.e., non-responders). Eleven patients (31%) had renal relapse after treatment for AIN.

Tumor responses to ICIs at the time of renal response assessment were as follows: 18 patients (51%) had tumor progression, 14 (40%) were in remission, and three (9%) had stable disease. Six patients (17%) had ICI re-challenge after treatment for AIN; five had AIN relapse and two had progression of their cancer.

### 3.2. Predictors of Renal Response

None of the baseline clinical characteristics evaluated, including comorbidities, exposure to nephrotoxic agents, use of nonsteroidal anti-inflammatory drugs or proton pump inhibitors, urine analysis findings, or the presence of acute tubular necrosis, were associated with renal response (Table 2). Pathologic features such as intensity of inflammation and tubulitis were not associated with renal response, but the presence of IFTA was associated with renal response (*p* = 0.02; Table 3). Higher numbers of CD8+ cells (*p* = 0.05), and CD8+ cell density (*p* = 0.07), were observed in responders. CD68 clustering was not observed in any of the samples.

### 3.3. OS and PFS

The median OS was not reached (NR) (95% confidence interval (CI) 35.22-NR). Responders had better 12-month OS rates than non-responders (77% (95% CI 56–89%) compared with 27% (95% CI 1–69%), *p* = 0.025; Figure 3a). Responders who received concurrent ICIs had the best 12-month OS rate (100%), responders who did not receive concurrent ICIs had a lower 12-month OS rate (72%, 95% CI 48–87%), and non-responders who did not receive concurrent ICIs had the worst 12-month OS rate (27%, 95% CI 1–69%; *p* = 0.041; Figure 3b).

The median PFS in the population was 13.1 months. Renal response was not associated with PFS (*p* = 0.11; Figure 4a), nor was use of concurrent ICIs (*p* = 0.27; Figure 4b).

## 4. Discussion

Our findings indicate that in patients with ICI-induced AIN, low levels of IFTA in kidney tissue specimens are associated with renal response to treatment for AIN within 3 months, and that renal response and use of concurrent ICIs are associated with better OS in these patients.

ICI-induced AIN is a rare irAE, and its incidence and predictors have been established primarily through retrospective analysis [2,4,12,13]. However, there is a paucity of data to support an evidence-based approach to the management and treatment of ICI-induced AIN, with guidelines based on expert opinion [18,19,20,21]. Several review papers have highlighted the importance of the kidney biopsy as a gold standard in diagnosing ICI induced nephritis, but evaluating the pathological features of AIN to help guide treatment further is yet to be elucidated [22].

The data on the use of a pathologic approach to evaluate irAE response has also been limited. Cortazar et al. reported that in 12 of 13 patients who developed AIN after treatment with ICIs, the infiltrates were composed predominantly of lymphocytes, with varying degrees of plasma cells and eosinophils. CD3 and CD20 staining revealed a predominance of CD3+ cells in only three patients [2]. The authors did not study biopsy findings and renal response, but they did note that their two patients with AIN who achieved a complete renal response to treatment of AIN had minimal to no fibrosis, whereas patients with no renal response had more fibrosis. Our findings confirm in a more robust statistical and pathologic analysis that the presence of high levels of fibrosis (i.e., IFTA) is a predictor of poor renal response. Therefore, the preservation of kidney function may be best achieved by early recognition of injury and the prompt treatment of ICI-induced AIN. This is further highlighted by our finding that patients who had renal response within 3 months of initiating treatment for AIN had better OS than non-responders.

Our study also evaluated immune subtypes CD4+, CD8+, CD20+, and CD68+. CD4+ and CD8+ cells are often found in irAEs in other organ systems, such as ICI-induced colitis, in which CD8+ cells are more predominant than in other inflammatory bowel disease [23]. The increased presence of B cells (CD20+) in irAEs and other autoimmune disease occurs as a result of activated T cell–B cell interaction, which can lead to autoantibody production. Therefore, the use of anti-CD20 rituximab (chimeric monoclonal antibody against the protein CD20) has been successfully used to treat hematologic, dermatologic, renal, neurologic, and rheumatologic irAEs [14,24,25,26,27]. Increasing evidence of CD68+ staining in myositis has been reported in studies showing that both CD4+/CD8+ and CD68+ macrophages are present in muscle biopsy specimens from patients with ICI-induced myositis and myasthenia gravis [28]. The current study did not show any association of renal response with the number of immune subtype cells or their density, clustering, or tubulitis, but we believe this is due to the small sample size.

An interesting aspect of ICI-induced AIN is its pathologic resemblance with kidney transplant rejection, in which lymphocyte-predominant tubulointerstitial infiltrate is present. This led us to evaluate our tissue specimens using the well-established Banff criteria for evaluation of kidney transplant rejection. The resemblance of ICI-induced AIN to kidney transplant rejection has been further validated in a study using NanoString technology, in which tissue samples from 15 patients with kidney transplant rejection were compared with those from 10 patients with ICI-induced AIN. In that study, almost all genes evaluated were not significantly differentially expressed between ICI-induced AIN and kidney transplant rejection, except for one gene, *IFI27*, which is inducible by interferon-alpha [29]. Similar to organ rejection, graft vs. host disease in liver tissue had similar immune expression to hepatis irAEs with infiltration of CD8+ T cells and defective accumulation of regulatory T cells expressing forkhead box p3 (FOXP3), but not in those from patients with autoimmune hepatitis [30].

Although our study is one of the largest detailed pathological evaluations of irAE induced nephritis we were limited in making more definitive associations, especially with the immune variables, due to the small sample size. However, what is evident is that there are features that defer in the biopsies such as presence or absence of eosinophils and granulomas, which suggest different pathways of injury. More work is needed to evaluate the pathophysiology of ICI induced nephritis and its distinct entity in comparison to drug induced nephritis, where simply stopping offending agent or treating with a short dose of steroids is not always effective. IrAE’s in other organ systems have been evaluated further and the use of steroid sparing agents has become the forefront of treatment in such scenarios, such as immune mediated colitis and arthritis [31,32]. In our experience, the use of infliximab has been successful in treating relapsed AIN induced by ICI [33]. The move towards steroid sparing agent has been as a result of concern to inhibiting tumor responses [34]. However, with the majority of the research in IrAE’s’s being clinical, there is a lack of understanding of the pathophysiology, genetic predisposition, and findings of the pathology, that would give a more guided approach and support the use of biologics in ICI induced nephritis.

## 5. Conclusions

Our detailed pathologic evaluation of ICI-induced AIN showed that the degree of fibrosis (i.e., IFTA) is associated with renal response within 3 months of initiating treatment, and that renal response and the use of concurrent ICIs was associated with better OS in these patients. Given the widespread use of ICIs in cancer treatment, further studies are needed to identify the novel biomarkers of risk for, and the effective early intervention and treatment of ICI-induced AIN, to improve both renal outcomes and OS in these patients.

## Figures and Tables

**Figure 1 cancers-14-05267-f001:**
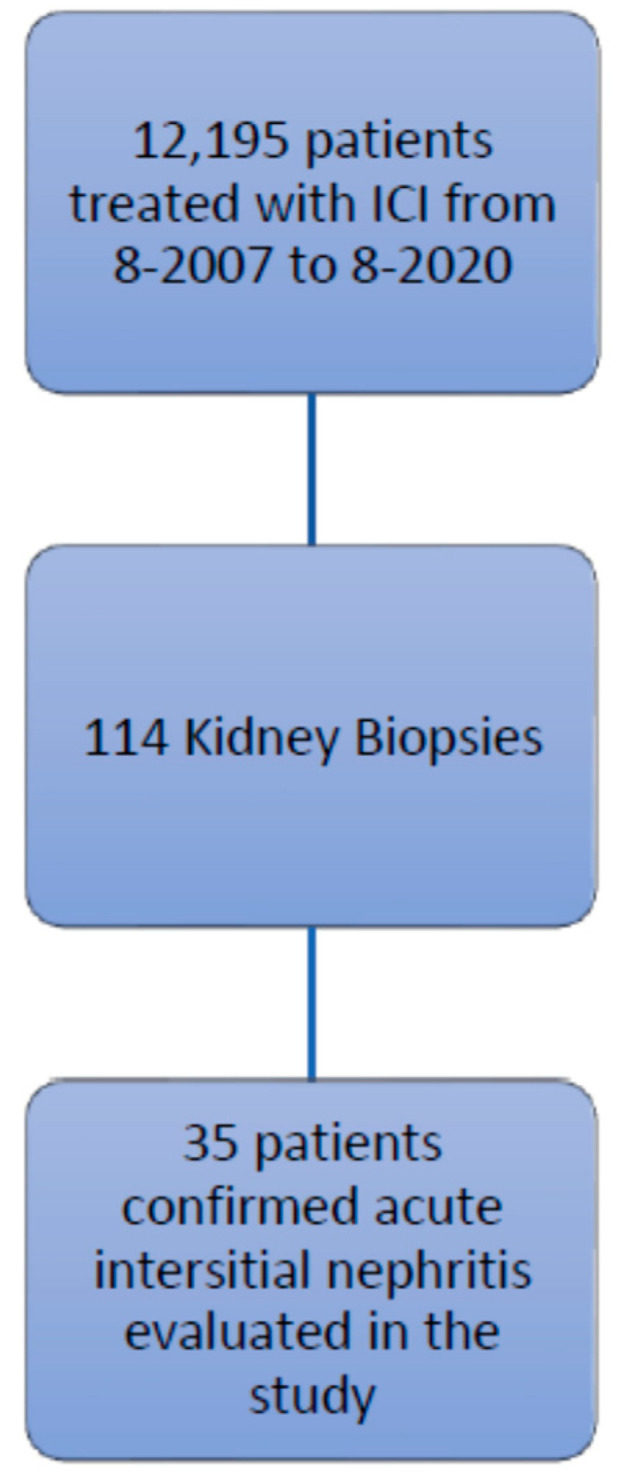
Study flow chart of inclusion criteria.

**Figure 2 cancers-14-05267-f002:**
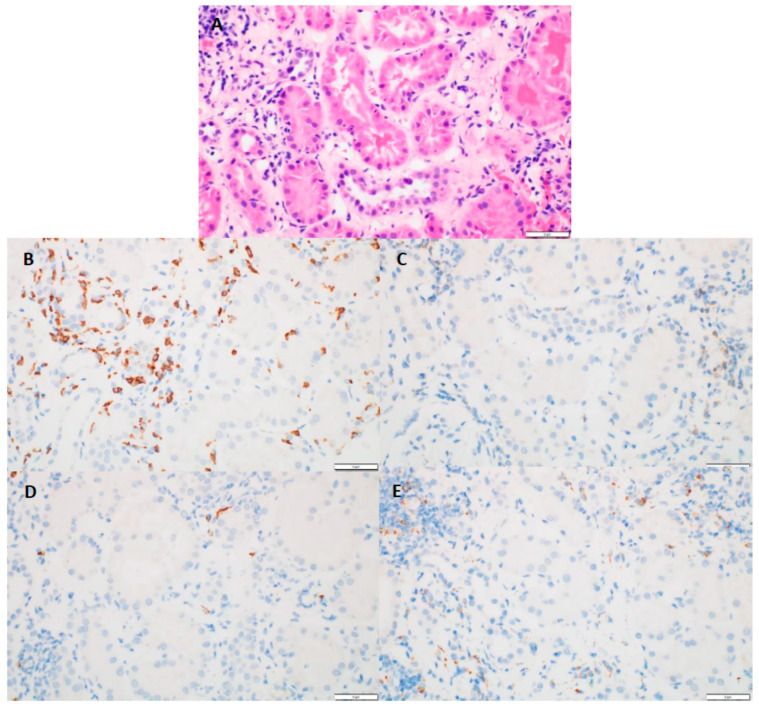
(**A**) (H&E, 40×)-tubulitis present in non-atrophic tubules; (**B**) (CD8, 40×)-numerous positive cells present with the areas of tubulitis; (**C**) (CD4, 40×)-absence of positive staining cells within tubulitis; (**D**) (C20, 40×)-absence of positive staining cells within tubulitis; (**E**) (CD68, 40×)-absence of positive staining cells within tubulitis.

**Figure 3 cancers-14-05267-f003:**
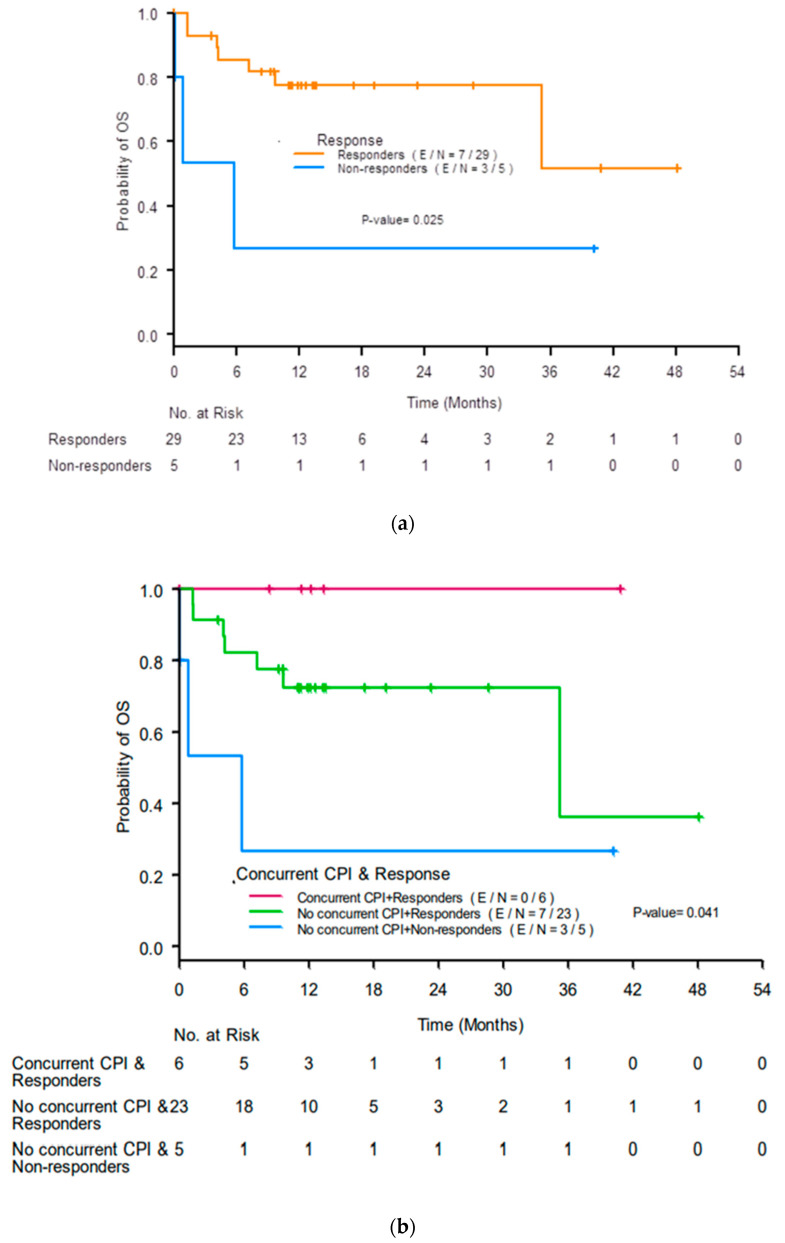
Kaplan-Meier curves for overall survival (OS) by (**a**) response to treatment for acute interstitial nephritis at 3 months after initiation of treatment and (**b**) renal response and use of concurrent immune checkpoint inhibitors (ICIs). CR, complete response; PR, partial response; NR, no response.

**Figure 4 cancers-14-05267-f004:**
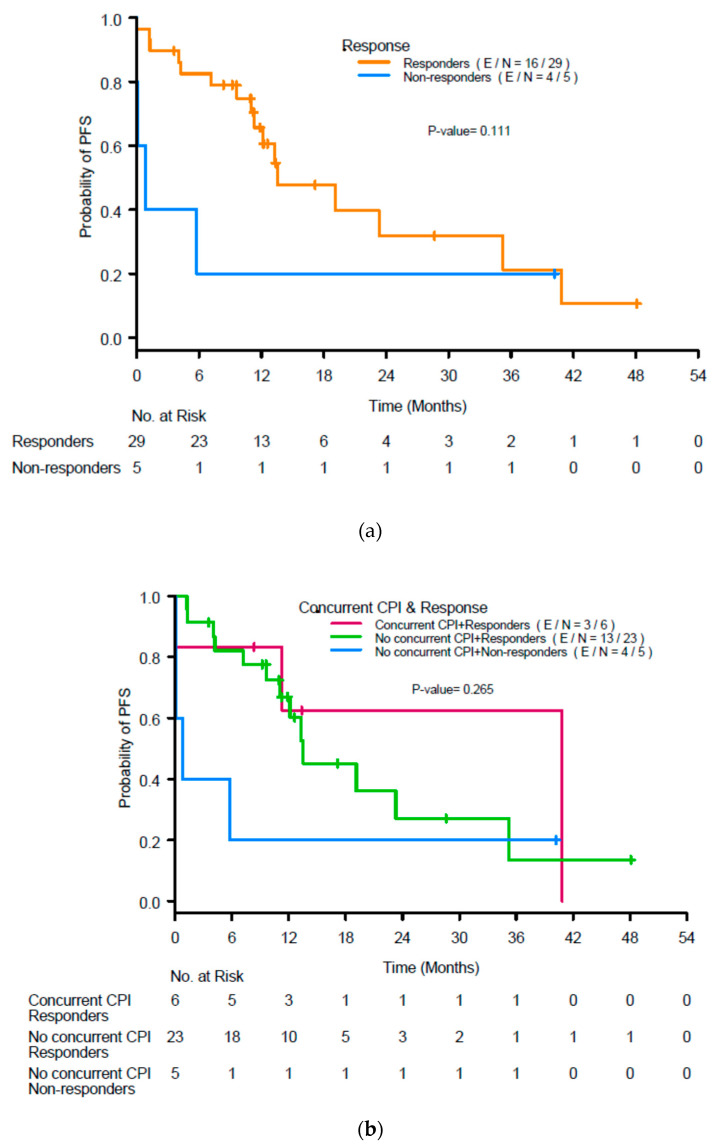
(**a**) Kaplan-Meier curves for progression-free survival (PFS) by response to treatment for acute interstitial nephritis at 3 months after initiation of treatment and (**b**) renal response and use of concurrent immune checkpoint inhibitors (ICIs). CR, complete response; PR, partial response; NR, no response.

**Table 1 cancers-14-05267-t001:** Baseline (i.e., at diagnosis of acute interstitial nephritis) demographic and clinical characteristics of patients who developed acute interstitial nephritis after treatment with immune checkpoint inhibitors (n = 35).

Variable	No. (%)
Sex	
Female	13(37)
Male	22 (63)
Hypertension	
No	15 (43)
Yes	20 (57)
Hyperlipidemia	
No	26 (74)
Yes	9 (26)
Diabetes mellitus	
No	28 (80)
Yes	7 (20)
Coronary artery disease	
No	34 (97)
Yes	1 (3)
Malignancy	
Breast cancer	1 (3)
Chronic lymphocytic leukemia	1 (3)
Colorectal cancer	1 (3)
Hodgkin lymphoma	2 (6)
Lung cancer	7 (20)
Non-Hodgkin lymphoma	1 (3)
Melanoma	10 (29)
Pancreatic cancer	1 (3)
Renal cell carcinoma	2 (6)
Renal cell carcinoma and chronic myelogenous leukemia	1 (3)
Rectal cancer	1 (3)
Smoldering myeloma	2 (6)
Thyroid cancer	1 (3)
Tonsil squamous cell carcinoma	1 (3)
Urothelial cancer	3 (9)
Immunotherapy	
Atezolizumab	3 (9)
Durvalumab	2 (6)
Nivolumab	17 (49)
Pembrolizumab	13 (37)
Concurrent nephrotoxic chemotherapy	
No	21 (60)
Yes	14 (40)
Concurrent nephrotoxic chemotherapy or immunotherapy	19 (54)
Use of concurrent immune checkpoint inhibitors	
No	29 (83)
Yes	6 (17)
Dialysis	
No	29 (83)
Yes	6 (17)
Steroids before biopsy	
No	25 (71)
Yes	10 (29)
Proton pump inhibitors or nonsteroidal anti-inflammatory drugs	
Nonsteroidal anti-inflammatory drug	1 (3)
Proton pump inhibitor	11 (31)
Both	1 (3)
None	22 (63)
Immune checkpoint inhibitor re-challenge	
No	29 (83)
Yes	6 (17)
Renal toxicity ^1^	
Grade II	13 (37)
Grade III	10 (29)
Grade IV	12 (34)
Red or white blood cells present in urine	
No	10 (29)
Yes	25 (71)
Other immune-related adverse events	
Arthritis	2 (6)
Colitis	3 (9)
Hepatitis	1 (3)
Hypothyroidism, myositis, and hypophysitis	1 (3)
Hypothyroidism	1 (3)
Neuritis	1 (3)
Pneumonitis	2 (6)
Pneumonitis and hepatitis	1 (3)
Pulmonary event and Sjogren syndrome	1 (3)
Rash	1 (3)
Uveitis	1 (3)
None	20 (57)
Biopsy results	
Chronic interstitial nephritis	
No	4 (11)
Yes	31 (89)
Acute tubular necrosis	
No	9 (26)
Yes	26 (74)
Granuloma	
No	30 (86)
Yes	5 (14)
Immunohistochemistry staining	
CD4	
Negative	10 (29)
Positive	25 (71)
CD4 clustering	
Negative	23 (66)
Positive	12 (34)
CD4 tubulitis	
Negative	27 (77)
Positive	8 (23)
CD8	
Negative	10 (29)
Positive	25 (71)
CD8 clustering	
Negative	16 (46)
Positive	19 (54)
CD8 tubulitis	
Negative	11 (31)
Positive	24 (69)
CD20	
Negative	11 (31)
Positive	24 (69)
CD20 clustering	
Negative	20 (57)
Positive	15 (43)
CD20 tubulitis	
Negative	34 (97)
Positive	1 (3)
CD68	
Negative	11 (31)
Positive	24 (69)
CD68 clustering	
Negative	35 (100)
Positive	0 (0)
CD68 tubulitis	
Negative	19 (54)
Positive	16 (46)

^1^ According to Common Terminology Criteria for Adverse Events.

**Table 2 cancers-14-05267-t002:** Association of baseline clinical characteristics with response to treatment for immune checkpoint inhibitor–induced acute interstitial nephritis (at 3 months after initiation of treatment).

Covariate	No. (% of Covariate Group)		*p*
	Responders, n = 29	Non-Responders, n = 6	
Sex			>0.99
Female	11 (84.6)	2 (15.4)	
Male	18 (81.8)	4 (18.2)	
Hypertension			0.37
No	11 (73.3)	4 (26.7)	
Yes	18 (90)	2 (10)	
Hyperlipidemia			0.31
No	20 (76.9)	6 (23.1)	
Yes	9 (100)	0 (0)	
Diabetes mellitus			>0.99
No	23 (82.1)	5 (17.9)	
Yes	6 (85.7)	1 (14.3)	
Coronary artery disease			>0.99
No	28 (82.4)	6 (17.6)	
Yes	1 (100)	0 (0)	
Immunotherapy			0.21
Atezolizumab	2 (66.7)	1 (33.3)	
Durvalumab	2 (100)	0 (0)	
Nivolumab	16 (94.1)	1 (5.9)	
Pembrolizumab	9 (69.2)	4 (30.8)	
Concurrent nephrotoxic chemotherapy			0.66
No	18 (85.7)	3 (14.3)	
Yes	11 (78.6)	3 (21.4)	
Concurrent immune checkpoint inhibitors			0.56
No	23 (79.3)	6 (20.7)	
Yes	6 (100)	0 (0)	
Dialysis			>0.99
No	24 (82.8)	5 (17.2)	
Yes	5 (83.3)	1 (16.7)	
Steroids before biopsy			0.65
No	20 (80)	5 (20)	
Yes	9 (90)	1 (10)	
Proton pump inhibitors or nonsteroidal anti-inflammatory drugs			0.38
Yes	12 (92.3)	1 (7.7)	
No	17 (77.3)	5 (22.7)	
Immune checkpoint inhibitor re-challenge			0.56
No	23 (79.3)	6 (20.7)	
Yes	6 (100)	0 (0)	
Renal toxicity			0.65
Grade III-IV	19 (86.4)	3 (13.6)	
Grade I-II	10 (76.9)	3 (23.1)	
Red or white blood cells present in urine			0.32
No	7 (70)	3 (30)	
Yes	22 (88)	3 (12)	
Chronic interstitial nephritis			0.55
No	3 (75)	1 (25)	
Yes	26 (83.9)	5 (16.1)	
Acute tubular necrosis			>0.99
No	8 (88.9)	1 (11.1)	
Yes	21 (80.8)	5 (19.2)	
Granuloma			0.56
No	24 (80)	6 (20)	
Yes	5 (100)	0 (0)	
Banff inflammation score			0.07
0	4 (80)	1 (20)	
1	6 (85.7)	1 (14.3)	
2	3 (50)	3 (50)	
3	16 (94.1)	1 (5.9)	
Low or high Banff inflammation score			>0.99
0–1	10 (83.3)	2 (16.7)	
2–3	19 (82.6)	4 (17.4)	
Maximum Banff inflammation score			0.18
0–2	13 (72.2)	5 (27.8)	
3	16 (94.1)	1 (5.9)	
Banff tubulitis score			0.13
0	5 (83.3)	1 (16.7)	
1	5 (62.5)	3 (37.5)	
2	7 (77.8)	2 (22.2)	
3	12 (100)	0 (0)	
CD4			>0.99
Negative	8 (80)	2 (20)	
Positive	21 (84)	4 (16)	
CD4 clustering			0.64
Negative	18 (78.3)	5 (21.7)	
Positive	11 (91.7)	1 (8.3)	
CD4 tubulitis			>0.99
Negative	22 (81.5)	5 (18.5)	
Positive	7 (87.5)	1 (12.5)	
CD8			>0.99
Negative	8 (80)	2 (20)	
Positive	21 (84)	4 (16)	
CD8 clustering			>0.99
Negative	13 (81.3)	3 (18.8)	
Positive	16 (84.2)	3 (15.8)	
CD8 tubulitis			0.35
Negative	8 (72.7)	3 (27.3)	
Positive	21 (87.5)	3 (12.5)	
CD20			>0.99
Negative	9 (81.8)	2 (18.2)	
Positive	20 (83.3)	4 (16.7)	
CD20 clustering			0.68
Negative	16 (80)	4 (20)	
Positive	13 (86.7)	2 (13.3)	
CD20 tubulitis			>0.99
Negative	28 (82.4)	6 (17.6)	
Positive	1 (100)	0 (0)	
CD68			>0.99
Negative	9 (81.8)	2 (18.2)	
Positive	20 (83.3)	4 (16.7)	
CD68 clustering			–
Negative	29 (82.9)	6 (17.1)	
Positive	0 (0)	0 (0)	
CD68 tubulitis			>0.99
Negative	16 (84.2)	3 (15.8)	
Positive	13 (81.3)	3 (18.8)	

**Table 3 cancers-14-05267-t003:** Association of pathologic markers with response (n = 26) or non-response (n = 9) to treatment for immune checkpoint inhibitor–induced acute interstitial nephritis at 3 months after initiation of treatment.

Marker	Response	No.	Mean	SD	SE	Min	Max	Median	Quartile 1	Quartile 3	*p*
Baseline creatinine mg/dL	Non-responders	6	1.10	0.29	0.12	0.76	1.50	1.03	0.90	1.37	0.57
	Responders	29	1.06	0.48	0.09	0.50	3.05	0.93	0.85	1.08	
Global glomerulosclerosis %	Non-responders	6	16.67	16.97	6.93	6.00	50.00	10.50	6.00	17.00	>0.99
	Responders	29	16.62	16.52	3.07	0.00	55.00	14.00	4.00	20.00	
Interstitial fibrosis with tubular atrophy %	Non-responders	5	33.00	13.04	5.83	20.00	50.00	35.00	20.00	40.00	0.02
	Responders	29	15.17	16.93	3.14	0.00	70.00	10.00	0.00	20.00	
Inflammation score	Non-responders	6	1.67	1.03	0.42	0.00	3.00	2.00	1.00	2.00	0.30
	Responders	29	2.07	1.16	0.22	0.00	3.00	3.00	1.00	3.00	
No. of eosinophils per 40× field	Non-responders	6	7.00	11.44	4.67	0.00	30.00	2.00	2.00	6.00	0.82
	Responders	29	8.83	20.61	3.83	0.00	101.00	2.00	0.00	5.00	
No. of neutrophils per 40× field	Non-responders	6	14.83	14.19	5.79	0.00	33.00	15.00	1.00	25.00	0.93
	Responders	29	17.76	24.57	4.56	0.00	89.00	7.00	2.00	28.00	
Peak creatinine	Non-responders	6	3.73	2.28	0.93	1.91	7.84	2.88	2.14	4.71	0.93
	Responders	29	3.63	2.05	0.38	1.46	9.57	2.96	2.26	4.83	
Tubulitis score	Non-responders	6	1.17	0.75	0.31	0.00	2.00	1.00	1.00	2.00	0.12
	Responders	29	1.90	1.14	0.21	0.00	3.00	2.00	1.00	3.00	
Final creatinine	Non-responders	5	3.00	1.94	0.87	1.91	6.43	2.00	1.97	2.69	0.003
	Responders	28	1.40	0.73	0.14	0.89	4.50	1.15	1.02	1.41	
No. of cycles	Non-responders	6	5.17	5.91	2.41	1.00	17.00	3.50	2.00	4.00	0.23
	Responders	29	10.76	22.12	4.11	1.00	123.00	7.00	3.00	8.00	
No. of CD20 cells	Non-responders	4	19.50	13.70	6.85	3	35	20.0	9.0	30.0	0.16
	Responders	21	40.67	29.81	6.50	0	115	33.0	19.0	54.0	
CD20 density %	Non-responders	4	25.50	20.42	10.21	2	50	25.0	10.0	41.0	0.22
	Responders	21	46.57	32.08	7.00	0	95	46.0	22.0	68.0	
CD20 (number per 10 epithelial cells)	Non-responders	4	0.00	0.00	0.00	0	0	0.0	0.0	0.0	0.74
	Responders	21	0.10	0.44	0.10	0	2	0.0	0.0	0.0	
No. of CD4 cells	Non-responders	4	46.00	44.37	22.18	19	112	26.5	20.0	72.0	0.63
	Responders	21	69.67	63.04	13.76	1	215	57.0	20.0	88.0	
CD4 density %	Non-responders	4	30.50	24.58	12.29	13	66	21.5	14.0	47.0	0.60
	Responders	21	45.90	35.35	7.71	3	140	45.0	13.0	70.0	
CD4 (number per 10 epithelial cells)	Non-responders	4	0.75	1.50	0.75	0	3	0.0	0.0	1.5	0.62
	Responders	21	1.48	2.25	0.49	0	6	0.0	0.0	3.0	
No. of CD68 cells	Non-responders	4	49.25	7.89	3.94	42	60	47.5	43.5	55.0	0.24
	Responders	21	87.43	71.55	15.61	0	220	70.0	30.0	85.0	
CD68 density %	Non-responders	4	35.50	15.84	7.92	21	58	31.5	25.5	45.5	0.50
	Responders	21	42.38	26.16	5.71	0	98	35.0	26.0	60.0	
CD68 (number per 10 epithelial cells)	Non-responders	4	3.00	2.45	1.22	0	5	3.5	1.0	5.0	0.97
	Responders	21	3.38	3.28	0.72	0	10	4.0	0.0	6.0	
No. of CD8 cells	Non-responders	4	83.50	32.56	16.28	35	105	97.0	65.5	101.5	0.05
	Responders	21	181.14	124.40	27.15	29	525	144.0	104.0	195.0	
CD8 density	Non-responders	4	46.75	21.76	10.88	17	69	50.5	32.5	61.0	0.07
	Responders	21	82.19	37.91	8.27	20	173	80.0	62.0	99.0	
CD8 (number per 10 epithelial cells)	Non-responders	4	4.50	3.42	1.71	0	8	5.0	2.0	7.0	0.33
	Responders	21	6.62	3.73	0.81	1	15	7.0	3.0	8.0	

## Data Availability

All data generated or analyzed during this study are included in this published article.

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
