# Peer review of "Pathologic Predictors of Response to Treatment of Immune Checkpoint Inhibitor–Induced Kidney Injury"

_cancers, 2022, doi:10.3390/cancers14215267_

Round 1

Reviewer 1 Report

A well-written paper discussing real-world experience.  Immune-related side effects are common and need close attention to etiology and intervention

Minor comments

-Most patients responded to steroids and had grade II-III toxicity. Please explain why re-challenging was done only in 17% of the patients.

- Did the occurrence of renal toxicity affect the survival? I think the authors can easily compare this with the patients who received ICI with developing nephritis.

-https://pubmed.ncbi.nlm.nih.gov/33912752/ à This paper discusses the ICI-AKI with non-ICI AKI. Can we include serum CRP and uRBP/Cr in our patient cohort?

- What was the median duration of these patients on steroid therapy?

-Please add a PRISMA diagram

Reviewer 2 Report

Author’s data pathologic evaluation of ICI-induced AIN showed that the degree of fibrosis (i.e., IFTA) is associated with renal response within 3 months of initiating treatment, and that renal response and use of concurrent ICIs was associated with better OS in these patients. Given the widespread use of ICIs in cancer treatment, further studies are needed to identify novel biomarkers of risk for and effective early intervention and treatment of ICI-induced AIN to improve both renal outcomes and OS in these patients.

This is a very meaningful study, and I have a few little suggestions:

First, as a general comment, we suggest that the authors increase the sample size in future studies.

In the result part, the resolution of the author's figures are not enough, and it will be distorted when enlarged, especially the texts on the figures are not clear.

In the discussion section, we suggest to further discuss the significance of this study compare with the current related research. In fact, the author's discussion section is short and cannot highlight the characteristics and advantages of the study.

Author Response

The figures have been created with higher resolution as a PDF file but unable to upload since system only allows one document to be uploaded.
